# Computational and Preclinical Prediction of the Antimicrobial Properties of an Agent Isolated from *Monodora myristica*: A Novel DNA Gyrase Inhibitor

**DOI:** 10.3390/molecules28041593

**Published:** 2023-02-07

**Authors:** Sunday Amos Onikanni, Bashir Lawal, Adewale Oluwaseun Fadaka, Oluwafemi Bakare, Ezekiel Adewole, Muhammad Taher, Junaidi Khotib, Deny Susanti, Babatunji Emmanuel Oyinloye, Basiru Olaitan Ajiboye, Oluwafemi Adeleke Ojo, Nicole Remaliah Samantha Sibuyi

**Affiliations:** 1College of Medicine, Graduate Institute of Biomedical Sciences, China Medical University, Taichung 40402, Taiwan; 2Biochemistry Unit, Department of Chemical Sciences, Afe Babalola University, Ado-Ekiti 360101, Nigeria; 3Department of Pathology, University of Pittsburgh, Pittsburgh, PA 15213, USA; 4Department of Biotechnology, University of the Western Cape, Bellville 7530, South Africa; 5Department of Biochemistry, Faculty of Science, Adekunle Ajasin University, Akungba Akoko 342111, Nigeria; 6Industrial Chemistry Unit, Department of Chemical Sciences, Afe Babalola University, Ado-Ekiti 360101, Nigeria; 7Department of Pharmaceutical Technology, Kulliyyah of Pharmacy, International Islamic University Malaysia, Kuantan 25200, Pahang, Malaysia; 8Pharmaceutics and Translational Research Group, Kulliyyah of Pharmacy, International Islamic University Malaysia, Kuantan 25200, Pahang, Malaysia; 9Department of Pharmacy Practice, Faculty of Pharmacy, Airlangga University, Surabaya 60115, Indonesia; 10Department of Chemistry, Kulliyyah of Science, International Islamic University Malaysia, Kuantan 25200, Pahang, Malaysia; 11Biotechnology and Structural Biology (BSB) Group, Department of Biochemistry and Microbiology, University of Zululand, Kwadlangezwa 3886, South Africa; 12Institute of Drug Research and Development, SE Bogoro Center, Afe Babalola University, PMB 5454, Ado-Ekiti 360001, Nigeria; 13Phytomedicine and Molecular Toxicology Research Laboratory, Department of Biochemistry, Federal University Oye-Ekiti, Oye-Ekiti 371104, Nigeria; 14Phytomedicine, Molecular Toxicology, and Computational Biochemistry Research Laboratory (PMTCB-RL), Department of Biochemistry, Bowen University, Iwo 232101, Nigeria; 15Health Platform, Advanced Materials Division, Mintek, Randburg 2194, South Africa

**Keywords:** Schrödinger, *Monodora myristica*, conformational, antibiotics, levofloxacin

## Abstract

The African nutmeg (*Monodora myristica*) is a medically useful plant. We, herein, aimed to critically examine whether bioactive compounds identified in the extracted oil of *Monodora myristica* could act as antimicrobial agents. To this end, we employed the Schrödinger platform as the computational tool to screen bioactive compounds identified in the oil of *Monodora myristica*. Our lead compound displayed the highest potency when compared with levofloxacin based on its binding affinity. The hit molecule was further subjected to an Absorption, Distribution, Metabolism, Excretion (ADME) prediction, and a Molecular Dynamics (MD) simulation was carried out on molecules with PubChem IDs 529885 and 175002 and on three standards (levofloxacin, cephalexin, and novobiocin). The MD analysis results demonstrated that two molecules are highly compact when compared to the native protein; thereby, this suggests that they could affect the protein on a structural and a functional level. The employed computational approach demonstrates that conformational changes occur in DNA gyrase after the binding of inhibitors; thereby, this resulted in structural and functional changes. These findings expand our knowledge on the inhibition of bacterial DNA gyrase and could pave the way for the discovery of new drugs for the treatment of multi-resistant bacterial infections.

## 1. Introduction

Bacterial infections cause more deaths in the developing nations than the developed nations [1]. A report commissioned by the UK government anticipates an increase in deaths caused by antimicrobial-resistance by 2050, with a prediction that 10 million people will fall victim to antibiotic-resistant infections annually: roughly 4.73 million in Asia and 4.15 million in Africa, in contrast to 0.39 and 0.32 million in Europe and the US, respectively [2]. Several countries identified as lower-middle income countries are more likely to lack access to clean water, be faced with sanitation and hygiene problems, and have greater numbers of their people be affected by bacterial pathogens. Moreover, the malnourished, immunocompromised, and HIV-positive citizens of these countries are particularly affected [2].

DNA gyrase, a member of the family of type IIA topoisomerases, is an essential protein in bacteria, as it introduces negative supercoils in their DNA via an ATP-dependent mechanism. The enzyme is an A2B2 tetramer consisting of GyrA and GyrB, and its inhibition has catastrophic effects on bacterial cell proliferation and survival; thus can serve as a major target for the development of antibacterial agents [3,4,5,6]. The DNA gyrase complex requires the transition between several conformational states to cleave duplex DNA, pass the DNA strand through the break, and relegate the DNA fragments [4,5,6,7].

Recently, several antibiotics such as levofloxacin have gained prominent importance in the market. However, bacterial resistance and their adverse effects (including dizziness, blunt vision, and headache) represent a serious concern. The re-evaluation of DNA gyrase as a target for the development of new antibiotics received considerable applause from researchers aiming to find a lasting solution to this menace. DNA gyrase inhibitors could act as effective agents against a broad range of pathogens, bind to new sites in these proteins, and inhibit their enzymes via a mechanism that is different from that of quinolones [8,9,10,11].

In recent years, a tremendous amount of research was conducted in order to explore the pharmacological utility of medicinal or plant-derived natural products [12]. Over the years, plants have been used for medicinal purposes by diverse people and cultures throughout the world, and many of them have been effective in treatment of various illness and diseases. The use of plants for medicinal purposes continues to this day, mostly in the form of traditional medicine, which is now recognized by the World Health Organization as a building block for primary health care [13,14,15].

African nutmeg (*Monodora myristica*), belongs to the family of Ananacea, and it is an important species of the evergreen forest of West Africa (mostly common in the Southern part of Nigeria) [16,17]. The seeds that are embedded in the white sweet-smelling pulp of the sub-spherical fruit of *Monodora myristica* are of particular economic importance, whereas the kernel when ground to powder is a popular condiment used for the preparation of pepper soup that serves as a stimulant for the relief from constipation and is also used to control the passive uterine hemorrhage that occurs immediately after birth [17,18]. 

Many bioactive compounds were identified in the oil extract of the plant as reported by Adewole et al. [12], including 1,3,3-trimethyl-2-oxabicyclo[2.2.2]octan-6-ol, called eucalyptol; it is a natural organic compound and a colorless liquid that is used as an ingredient in many brands of mouthwash and cough suppressants. Eucalyptol is known to control the hypersecretion of airway mucus and asthma, as it can suppress arachidonic acid metabolism and cytokine production in human monocytes [19,20]. It was also found to possess antibiotic activities [21].

The therapeutic potentials of fatty acids (FAs) were found to be particularly promising for the treatment of various microbial infections. The antibacterial activity of FAs is well-known in the literature and represents a promising option for the development of the next generation of antibacterial agents [22]. Moreover, FAs are highly involved in living organisms’ defense systems against numerous pathogens, including multidrug-resistant bacteria. Previous studies established a computational approach to the investigation of protein–ligand interactions and their stability in relation to the structure and the function of a receptor [22]. We previously identified hit molecules from plant extracts that are theoretically capable of protein inhibition [23,24,25,26,27]. Herein, we employed computational prediction tools in order to study the binding energy and interaction of these ligands against DNA gyrase. Finally, a Molecular Dynamics (MD) simulation was undertaken in order to investigate the stability of the interaction over a period of 100 ns. Therefore, this study utilized the characterized secondary metabolites deriving from *Monodora myristica*-extracted oil as potential antimicrobial agents that could suppress the bacterial cell growth, and it employed a computational approach to predict alternative compounds with drug-like properties and better inhibitory activities.

## 2. Results

The estimated number of hydrogen bonds that would be donated by the solute to water molecules in an aqueous solution is proportional to the number of hydrogen bonds that will be accepted by the solute when they interact with water molecules. Chemical structures of eucalyptol when compared with standard ligands are shown in Figure 1. The eucalyptol showed a successfully processed molecule on predicted central nervous system activity of +1 as shown Table 1.

Therefore, with the crucial analysis carried out on the efficacy of the molecules, all of the examined parameters were within Lipinski’s rule of five (ROF) cut-off range for the test compound 1,3,3-trimethyl-2-oxabicyclo[2.2.2]octan-6-ol and showed no violation of Lipinski’s ROF and Veber rules as shown in Table 2.

Furthermore, Table 3 shows that the ligand of interest (529885) exerted no inhibition upon the Pgp substrate when compared with cephalexin and levofloxacin, which were predicted to be inhibitors of the Pgp substrate. This showed that the ligand revealed an ATP-dependent drug efflux pump for xenobiotic compounds with broad substrate specificity.

As highlighted in Table 3, the qualitative human oral absorption of the DNA gyrase subunit A N-terminal domain and the *E. coli* DNA gyrase–DNA binding and cleavage domain in State 1 receptor complex with 1,3,3-trimethyl-2-oxabicyclo[2.2.2]octan-6-ol was of preference in different oral absorption routes when compared with the DNA gyrase subunit A N-terminal domain and *E. coli* DNA gyrase–DNA binding and cleavage domain in State 1 receptor complex with cephalexin and levofloxacin. Aside from that, based on a quantitative multiple linear regression model, the predicted human oral absorption displayed a very high percentage when compared against that of the control ligands (Table 3).

The medicinal chemistry and drug likeness outputs from our experiments with DNA gyrase subunit A N-terminal domain and *E. coli* DNA gyrase–DNA binding and cleavage domain in State 1 receptor complex with 1,3,3-trimethyl-2-oxabicyclo[2.2.2]octan-6-ol produced excellent results when compared with those of cephalexin and levofloxacin (Table 4).

The docking analysis approach already revealed several suitable drug molecules for the target of interest [28,29,30]. This was also the case after undertaking a similar approach with the DNA gyrase subunit A N-terminal domain and the *E. coli* DNA gyrase–DNA binding and cleavage domain in State 1 receptor complex with 1,3,3-trimethyl-2-oxabicyclo[2.2.2]octan-6-ol, producing a competitive docking score (−4.115 kcal/mol). At the same time, the docking scores for cephalexin and levofloxacin were −6.13 kcal/mol and −4.658 kcal/mol, respectively (Table 5). This same inhibitory effect was demonstrated with the highest level of glide ligand efficiency, and it is indicative of a more suitable binding target of interest against the control ligands.

Furthermore, based on specific interactions, the ARD98 in the DNA gyrase subunit A N-terminal domain receptor was involved in hydrogen bonding with the atoms showing no π-π interaction or salt bridge formed with 1,3,3-trimethyl-2-oxabicyclo[2.2.2]octan-6-ol, whereas GLY120, ARG98, SER118, and THR230 were involved in hydrogen bonding in the DNA gyrase subunit A N-terminal domain receptor with atoms in cephalexin and levofloxacin. Cephalexin had ARG98 at the salt bridge on interaction, whereas levofloxacin had no atoms at the salt bridge interaction. Moreover, GLU381 at the salt bridge interacted with levofloxacin but had no interaction at the salt bridge of cephalexin. On the other hand, VAL420 and GLU381 were involved in *E. coli* DNA gyrase–DNA binding and cleavage domain in State 1 receptor hydrogen bonding with the atoms in cephalexin and levofloxacin, whereas ALA421 and VAL420 were involved in the hydrogen bonding with atoms with no π–π interaction or salt bridge formed in 1,3,3-trimethyl-2-oxabicyclo[2.2.2]octan-6-ol, as shown in Table 5. This is an indication of good measurable binding affinities for receptor residues with ligand contribution to the flexibility of the target.

The molecular analysis undertaken on the 3D and 2D structures of the complexes formed by the top three compounds (i.e., 1,3,3-trimethyl-2-oxabicyclo[2.2.2]octan-6-ol acetate, cephalexin, and levofloxacin) with the target proteins revealed that the ligand of interest can occupy the active site of the enzyme (Figure 2, Figure 3 and Figure 4).

The three compounds, like the standard ligands, interacted with the DNA gyrase subunit A N-terminal domain at the active site amino acid residues SER118, THR230, and ARG98, respectively, and they also interacted with the *E. coli* DNA gyrase–DNA binding and cleavage domain in State 1 at the active site amino acid residues ALA421, VAL420, and HIS471, respectively (Figure 5, Figure 6 and Figure 7).

The dynamic behavior of the protein residues was further studied by examining the RMSF pattern and by calculating the mean fluctuations of the Apo protein in complex with the standard drugs and the studied molecules (Table 6). In agreement with the RMSD results, the fluctuation patterns of the complex-forming molecules were relatively the same as that of the Apo protein. The highest mean fluctuation was observed when the Apo protein was complexed with levofloxacin (1.57 Å), followed by 1KIJ–529885 (1.35 Å). The Apo protein had the smallest mean fluctuation of 1.29 Å, followed by 1KIJ–27447 (1.33 Å). For the Apo protein, there were five amino acid residues: GLU84 (2.45 Å), GLY100 (2.65 Å), ARG161 (2.10 Å), GLU305 (2.64 Å), and GLN335 (2.49 Å). The residue positions and the distance of the fluctuation between the complexes include: (i) GLU84 (2.27 Å), SER99 (307 Å), LYS306 (3.53 Å), and GN335 (3.02 Å) in the case of the 1KIJ–levofloxacin complex, (ii) GLN105 (2.53 Å). ARG161 (211 Å), HIS2110 (2.24 Å), LYS306 (2.53 Å), and GLN335 (3.17 Å) in the case of the 1KIJ–27447 complex, and (iii) GLU84 (2.96 Å), GLY101 (3.14 Å), GLN105 (2.69 Å), LYS306 (2.96 Å), and GLN335 (2.63 Å) in the case of the 1KIJ–529885 complex. Only fluctuations above 2 Å were herein reported, except for the C and N-terminal residues.

## 3. Discussion

Despite the important role of bacteria in the environment in which we live, bacterial infections exert a large impact on public health across the globe [31]. Although bacterial infections are considered to be easier to treat than viral infections, the resistance of the bacteria to several antimicrobial agents is becoming a growing concern with potentially devastating consequences [31]. To date, attention is mainly directed toward the development of small molecules for the targeting of bacterial infections; however, the current clinical pipeline is insufficient in tackling the sporadic emergence and spread of antimicrobial resistance [32]. Therefore, we herein employed computational screening methods in order to assess the therapeutic potential of secondary metabolites deriving from *Monodora myristica*-extracted oil as antimicrobial agents [12]. The undertaken computational analysis revealed that 1,3,3-trimethyl-2-oxabicyclo[2.2.2]octan-6-ol can display a better binding affinity when compared with approved antimicrobial agents, and it is characterized by a competitive docking score and glide ligand efficiency against levofloxacin (as shown in Table 6). The inhibitory potential of the protein–ligand complex was revealed by the docking score (Table 6) [33]. It is known that ligands are predicted to have a hydrogen bond donor score ≤5 and a hydrogen bond acceptor score ≤10 with successful processed molecular structures on a predicted central nervous system activity of +2. Therefore, the ligand of interest showed a successfully processed molecule on a predicted central nervous system activity of +1. An attempt to change any of the single molecular properties of the aforementioned could result in absorption and bioavailability [34]. However, the main test ligand (1,3,3-trimethyl-2-oxabicyclo[2.2.2]octan-6-ol) exhibited the most desirable pharmacological potential, with the parameters fitting within the ROF cut-off range and not violating the rule.

Furthermore, the ligand 1,3,3-trimethyl-2-oxabicyclo[2.2.2]octan-6-ol exhibited a good metabolic profile with the Pgp substrate, and thereby, it showed no inhibitory effects on the enzyme’s activity when compared with those of standard antimicrobial agents cephalexin and levofloxacin (Table 2). Moreover, the blood–brain barrier’s (BBB) permeant inhibitory property of the target ligand 1,3,3-trimethyl-2-oxabicyclo[2.2.2]octan-6-ol was increased, whereas it was not inhibited in the case of cephalexin. The potentiality of any drug candidate is traceable to oral bioavailability with a uniqueness in its functions, and it would enable the drug to pass through a cellular membrane. Otherwise, the drug could easily be trapped within these barriers, thereby posing a serious health challenge. Among these functions are desolvation, diffusion, resolvation, and physicochemical properties such as lipophilicity [35]. The context of the absorption, distribution, and excretion of drugs provides several ways to identify pharmacokinetic models, such as human intestinal absorption, aqueous solubility (absorption), p-glycoprotein inhibition (distribution), and renal organic cationic transporters inhibition (excretion). Therefore, it is important to identify the aforementioned pharmacokinetic parameters, as they provide additional knowledge for drug acceptability or rejection in any drug development process [36]. The hit molecule exhibited its permeability effect in the human intestinal membrane and the BBB.

Furthermore, in an attempt to validate the pharmacological potential of the hit compound, the native ligand 1,3,3-trimethyl-2-oxabicyclo[2.2.2]octan-6-ol was docked with the drug targets, and the intermolecular interactions with most of the crucial amino acid residues were revealed (Table 6); thereby, it suggests an interaction of the amino acids with the ligand at the binding site, which could infer that the selected hit served as an antimicrobial agent.

Moreover, MD simulation studies were performed for 100 ns to analyze the atomic level changes in DNA gyrase with respect to the timescale. In order to gain insight into the effect of the selected compounds as efficient inhibitors of DNA gyrase, their stability in the protein’s active site was investigated by using Schrödinger version 2022-1 to conduct the MD simulation on the native protein and in complex with the compounds. MD properties including the root mean square deviation (RMSD), the root mean square fluctuation (RMSF), the radius of gyration (rGyr), and the solvent-accessibility surface area (SASA) were analyzed.

Furthermore, the Analysis of the best-predicted docking pose interaction as shown in (Figure 8) and (Figure 9) revealed *E. coli* DNA gyrase–DNA binding domain with 1,3,3-trimethyl-2-oxabicyclo[2.2.2]octan-6-ol complexed with 1KIJ–cephalexin, as superimposed with the standard. Also, the RMSD and the RMSF were plotted in order to examine the stability (Figure 10A) and the flexibility (Figure 10B) of the Apo protein compared with its interaction with inhibitory molecules. The rGyr and the SASA were plotted in order to analyze the compactness (Figure 10C) and to compute the secondary structure of the complexes (Figure 10D).

The structural stability of the protein alone and in complex with the molecules was interpreted by the deviation of the plot and the mean ± SEM (Table 6). The deviation patterns of the each of these complexes and of the Apo protein were relatively the same throughout the simulation time of 100 ns. The RMSD alteration was observed in all of the RMSD patterns at 30–60 ns, after which they all attained an equilibrium until the end of the simulation period. By comparing the mean ± SEM of the plots, the 1KIJ–529885 complex exhibited the lowest RMSD value (2.87 Å), whereas the 1KIJ–27447 complex (3.89 Å) appeared to be higher when compared to the Apo protein (3.16 Å) and the 1KIJ–levofloxacin complex (3.38 Å). The differences in mean deviation explain the impact of the interaction of the molecules in the binding pocket of the receptor, and thus, provide a basis for further investigation.

Subsequently, the rGyr for the complexes were analyzed in order to deduce their changes in compactness over the 100-ns simulation period (Figure 10C and Table 5). This plot shows that the 1KIJ–529885 complex had the smallest rGyr mean value of 2.13 Å, and it was stable from the onset and throughout the 100-ns simulation period. The 1KIJ–27447 complex exhibited a mean rGyr value of 3.85 Å, and its pattern fluctuation was observed at the beginning of the simulation time until 30 ns, after which there were no observable changes. The 1KIJ–levofloxacin complex had the highest mean rGyr of 3.97 Å; however, a noticeable drop in its pattern was observed at 60 ns, but the pattern was stable until 90 ns, after which it returned to its original position until the end of the simulation. Taken together, the two molecules displayed a slight pattern of deviation and fluctuation when compared to the Apo protein and the standard complex. Furthermore, these complexes showed even greater compactness than that exhibited by the standard (levofloxacin). All of these results demonstrate that the studied molecules formed highly stable complexes with DNA gyrase.

In addition, the SASA examined the interaction between the protein surface and surrounding solvent molecules. The interactions during the protein folding may be crucial to its stability and rearrangement with respect to its structure. As such, the SASA values of the complexes with respect to the standard were computed (Figure 10D and Table 5). The standard complex had a mean SASA value of 72.78 Å, whereas both the 1KIJ–27447 and the 1KIJ–529885 had higher mean SASA values of 127.30 Å and 150.99 Å, respectively. The observed SASA fluctuation is associated with the rearrangement of the amino acid residues from either the accessible region or the buried region. This may contribute to alterations in the protein–ligand overall structure.

## 4. Materials and Methods

The computational tools used for this study were part of the Schrödinger Platform (version 2020-3 for Windows 10).

### 4.1. Preparation of Ligands

A library of ten small molecular weight compounds was used in this in silico study, which was based on the bioactive compounds we previously identified and isolated from the oil of *Monodora myristica* [12]. The preparation of the ligands was carried out by using the Ligprep module of Maestro 20.3 with an OPLS3 force field at pH 7.0 ± 2.0 [37]. The “desalt” and “generate tautomers” options were selected, and the stereoisomer computation was left to generate at most 32 binding sites per ligand. The output format was left as that of Maestro’s defaults.

### 4.2. Preparation of Proteins

The proteins of interest (Protein Data Bank IDs: 3ILW, 1KIJ, and 6RKU) were retrieved from the Research Collaboratory for Structural Bioinformatics database (http://www.rcsb.org/pdb accessed on 2 November 2022) and were uploaded to the workspace of Maestro 20.3. The proteins were prepared via the protein preparation wizard of the Schrödinger platform. During the pre-processing of the proteins, bond orders were assigned, water molecules were deleted within 2.8 Å from het groups, and het states were set at pH 7.0 ± 2.0 [38]. Subsequently, hydrogen bonds were added, and ions were removed. In the “Refine” tab, the H-bond network was optimized by using PROPKA, and water molecules with less than three H-bonds to non-water molecules were removed [39]. Energy minimization was carried out by using the OPLS3 force field with an RMSD at 0.30 Å.

### 4.3. Receptor Grid Generation

The grid file receptor was generated by using the receptor grid generation panel that represents the active sites of the receptor for the simulation of glide ligand docking. The ligand-binding site was defined by the co-crystalized ligand within the workspace or the sitemap, a module of the Schrödinger platform. The van der Waals radii of the receptor atoms with partial atomic charge were set at a scaling factor of 1.0, and a partial cut-off of 0.25 was applied in order to soften the potential for the nonpolar parts of the receptor.

### 4.4. Molecular Docking Analysis

The compounds prepared from PubChem were docked into the active sites of the protein by using extra precision with the ligand sampling set as “non-refined”. Prior to the docking of the prepared compounds with said protein, the co-crystallized ligand was docked into the active site of the protein to predict binding affinity and molecular interaction.

### 4.5. Predictions of ADME Properties and Toxicological Potential

The Absorption, Distribution, Metabolism, Excretion (ADME), the toxicity of the compounds, and the molecular properties of the lead compounds were predicted by using the Qikprop, the Maestro 20.3, and the AdmetSAR 2.0 web-based tools, respectively [40].

### 4.6. Binding Free Energy Calculation

The free energy binding of the protein-ligand complexes was accessed in order to determine the stability of these complexes via the Prime MM-GBSA program (Schrödinger suite Maestro 20.3). The Prime MM-GBSA panel was used in order to calculate the binding free energy for the ligands’ protein complexes by using the MM-GBSA technology available with Prime [41,42]. The OPLS3 force field was selected, and VSGB was used as the continuum solvent model. All other options were set as default.

### 4.7. MD Simulations and Trajectory Analysis

The MD simulation for the native protein (1KIJ) and the three complexes (1KIJ–levofloxacin, 1KIJ–27447, and 1KIJ–529885) was carried by using Schrödinger 2022 version 1 with Maestro version 13.1.137, MM share version 5.7.137, and Windows-x64 Platform. The system setup and the MD preparation and trajectory analysis methods were similar to those of a previous method of ours, with slight modification [28,29]. Briefly, the docked complexes were individually subjected to MD by using the Desmond module of the Schrödinger software with an OPLS 2005 force field. The protein–ligand complex was bounded with a predefined transferable intermolecular potential with a 3-point water model in an orthorhombic box. The volume of the box was minimized, and the overall charge of the system was neutralized by adding sodium and chloride ions to mimic physiological conditions. The temperature and the pressure were kept constant at 310 Kelvin and 1.01325 bar, respectively, by using a Nose-Hoover thermostat and a Martyna-Tobias-Klein barostat made from United State. The simulation relaxation was undertaken by using an NPT ensemble after considering the number of atoms, the pressure, and the timescale. During the MD simulation, the long-range electrostatic interactions were calculated by using the particle mesh Ewald method. In addition, the MD simulation was carried out for 100 ns, and the trajectory sampling was set at an interval of 100 ps with 1000 frame numbers. Simulation outputs were analyzed and visualized by a simulation interaction diagram and an MS-MD trajectory analysis. The MD analysis was done in replicate to avoid variation. Data were plotted by using OriginPro version 9.

## 5. Conclusions

The resistance of bacteria to several antimicrobial agents has become an issue of major concern with potentially devastating consequences if not approached seriously. The herein identified hit candidate displayed satisfactory pharmacokinetic properties, exhibited interesting binding affinities, and did not violate Lipinski’s ROF; thereby, the results indicate a safe treatment option for bacterial infections. Moreover, our computational analyses provide the theoretical inhibitory activities of the identified molecules against DNA gyrase.

## Figures and Tables

**Figure 1 molecules-28-01593-f001:**
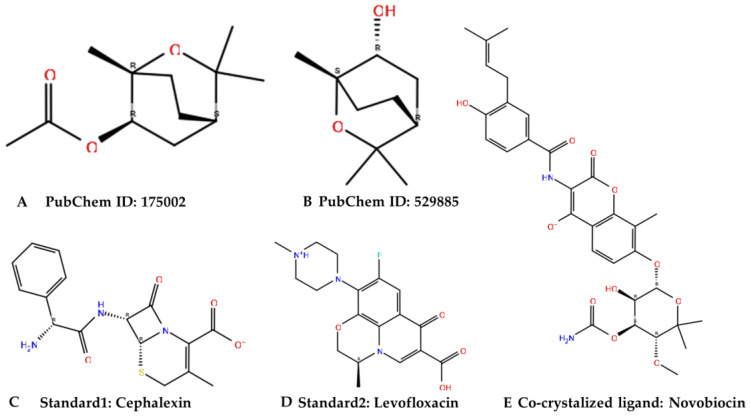
Chemical structures of the identified active molecules (**A**,**B**) with cephalexin (**C**) and levofloxacin (**D**) as standards and novobiocin (**E**) as a co-crystalized ligand were shown in Figure 1 above.

**Figure 2 molecules-28-01593-f002:**
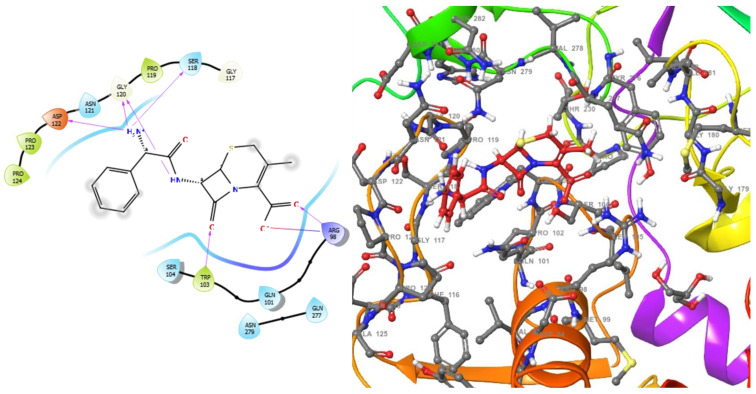
2D (**left**) and 3D (**right**) views of the molecular interactions at the surface of the active site of the DNA gyrase subunit A N-terminal domain binding with cephalexin (the green stick model). The negative, positive, and neutral charges of the binding site residues are represented with red, blue, and white colors, respectively.

**Figure 3 molecules-28-01593-f003:**
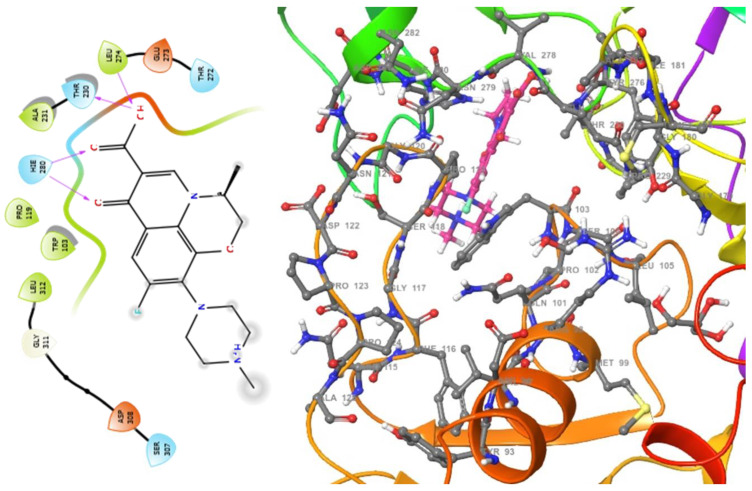
2D (**left**) and 3D (**right**) views of the molecular interactions at the surface of the active site of the DNA gyrase subunit A N-terminal domain binding with levofloxacin (the green stick model). The negative, positive, and neutral charges of the binding site residues are represented with red, blue, and white colors, respectively.

**Figure 4 molecules-28-01593-f004:**
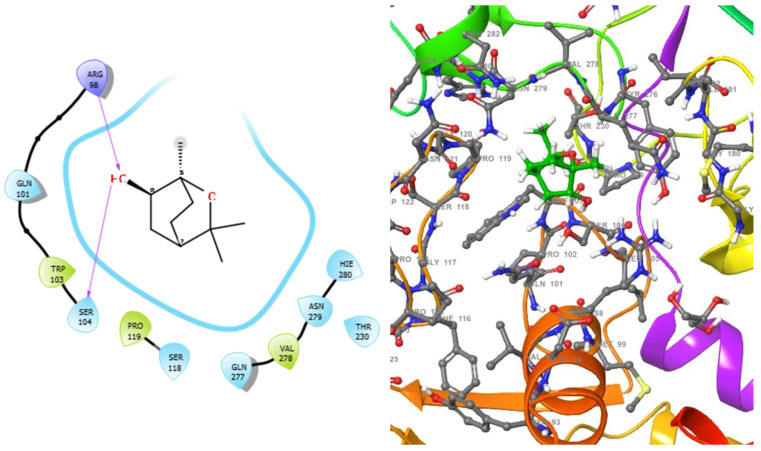
2D (**left**) and 3D (**right**) views of the molecular interactions at the surface of the active site of DNA gyrase subunit A N-terminal domain binding with 1,3,3-trimethyl-2-oxabicyclo[2.2.2]octan-6-ol (the green stick model). The negative, positive, and neutral charges of the binding site residues are represented with red, blue, and white colors, respectively.

**Figure 5 molecules-28-01593-f005:**
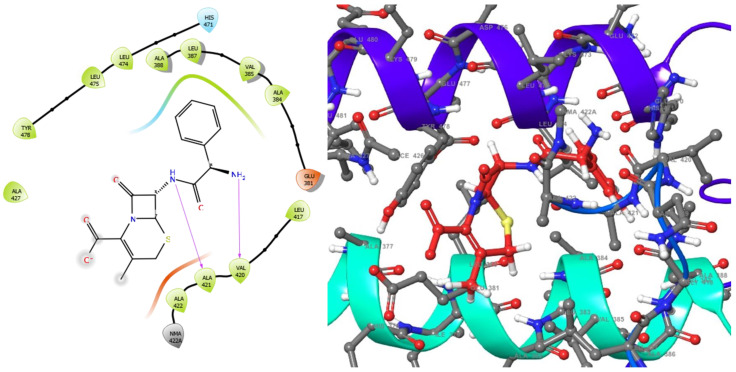
2D (**left**) and 3D (**right**) views of the molecular interactions at the surface of the active site of the *E. coli* DNA gyrase–DNA binding and cleavage domain in State 1 after binding with cephalexin (the green stick model). The negative, positive, and neutral charges of the binding site residues are represented with red, blue, and white colors, respectively.

**Figure 6 molecules-28-01593-f006:**
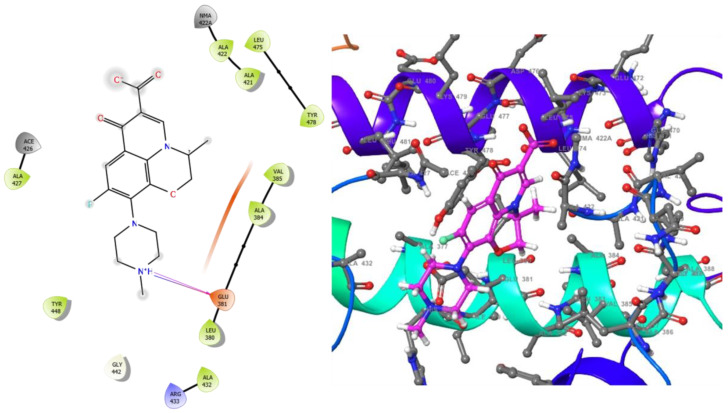
2D (**left**) and 3D (**right**) views of the molecular interactions at the surface of the active site of *E. coli* DNA gyrase–DNA binding and cleavage domain in State 1 binding with levofloxacin (the green stick model). The negative, positive, and neutral charges of the binding site residues are represented with red, blue, and white colors, respectively.

**Figure 7 molecules-28-01593-f007:**
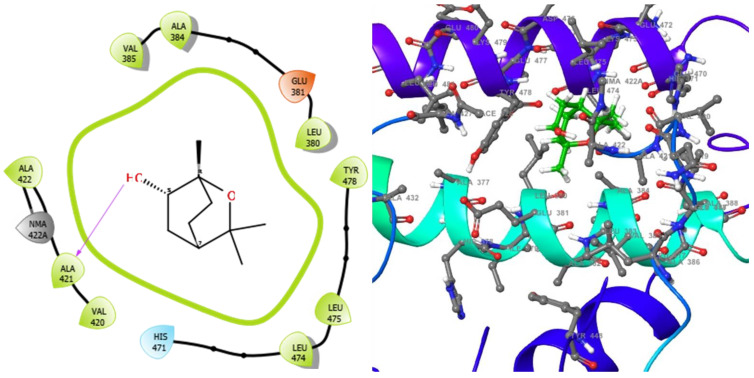
2D (**left**) and 3D (**right**) views of the molecular interactions at the surface of the active site of receptor *E. coli* DNA gyrase–DNA binding and cleavage domain in State 1 binding with 1,3,3-trimethyl-2-oxabicyclo[2.2.2]octan-6-ol (the green stick model). The negative, positive, and neutral charges of the binding site residues are represented with red, blue, and white colors, respectively.

**Figure 8 molecules-28-01593-f008:**
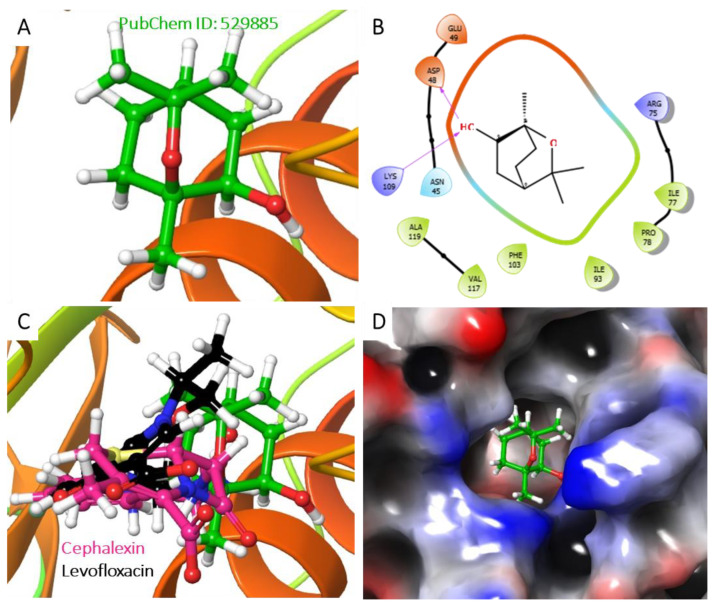
Analysis of the best-predicted docking pose of 529885. (**A**) 3D representation of the 1KIJ–cephalexin complex. (**B**) 2D interaction diagram of the 1KIJ–cephalexin complex. (**C**) 3D representation of the 1KIJ–cephalexin complex, as superimposed with the standard levofloxacin (green backbone). (**D**) Hydrophobic surface representation of 1KIJ when complexed with cephalexin as shown in Figure 8.

**Figure 9 molecules-28-01593-f009:**
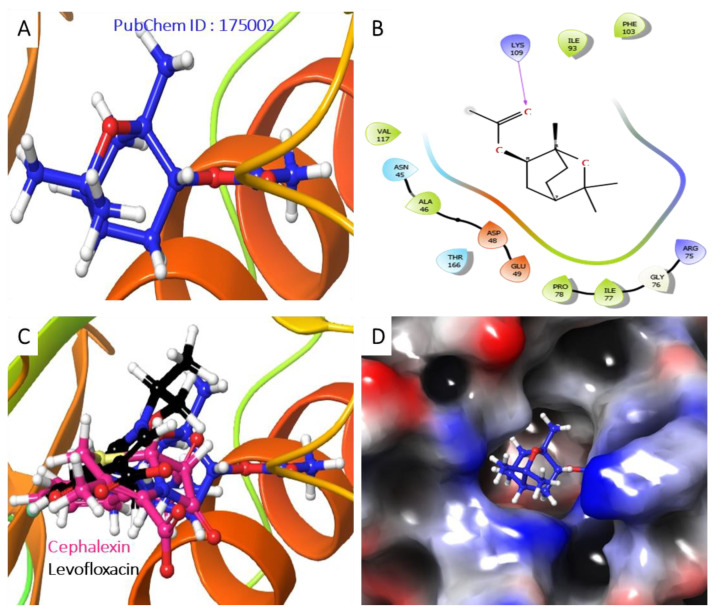
Analysis of the best-predicted docking pose of 175002. (**A**) 3D representation of the 1KIJ–1,3,3-trimethyl-2-oxabicyclo[2.2.2]octan-6-ol complex. (**B**) 2D interaction diagram of the 1KIJ–1,3,3-trimethyl-2-oxabicyclo[2.2.2]octan-6-ol complex. (**C**) 3D representation of the 1KIJ–1,3,3-trimethyl-2-oxabicyclo[2.2.2]octan-6-ol complex as superimposed with standard levofloxacin (green backbone). (**D**) Hydrophobic surface representation of 1KIJ when complexed with 1,3,3-trimethyl-2-oxabicyclo[2.2.2]octan-6-ol as shown in Figure 9.

**Figure 10 molecules-28-01593-f010:**
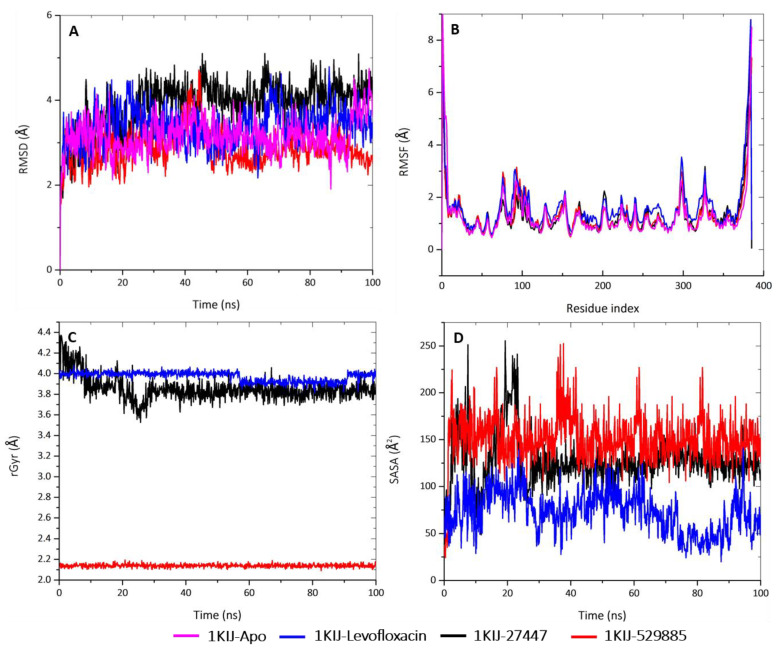
MD analyses of Apo-1KIJ and 1KIJ complexed to levofloxacin, cephalexin, and 1,3,3-trimethyl-2-oxabicyclo[2.2.2]octan-6-ol. (**A**) RMSD graphical illustration, (**B**) RMSF plot, (**C**) rGyr representation, and (**D**) SASA diagram. All MD simulations were performed by using Schrödinger version 2022_1.

**Table 1 molecules-28-01593-t001:** Physicochemical properties of the ligands docked with the target proteins.

PubChem ID	MW	CNS	donorHB	accptHB	dip^2/V	# Acid
529885	170.251	+1	1	2.45	0.009151	0
27447	347.388	−2	3.25	7.25	0.080638	1
175002	212.288	+1	1	0.75	0.012765	0
149096	361.372	0	0	0.75	0.004096	1
Co-Lig	612.632	−2	5.25	13.15	0.018557	0

Abbreviations used: accptHB: acceptor hydrogen bond; CNS: central nervous system; dip^2/V: square of the dipole moment divided by molecular volume; donorHB: donor hydrogen bond; MW: molecular weight; # Acid: number of carboxylic acids; 529885: (1,3,3-trimethyl-2-oxabicyclo[2.2.2]octan-6-ol; 27447: cephalexin; 175002:1,3,3-trimethyl-2-oxabicyclo[2.2.2]octa-6-yl) acetate; 149096: levofloxacin.

**Table 2 molecules-28-01593-t002:** Bio-absorbability output of test compounds.

PubChem ID	HOA	% HOA	SAfluorine	ROF	ROT	PSA
529885	3	100	0	0	0	29.867
27447	2	31.065	0	0	1	138.824
175002	3	100	0	0	0	39.375
149096	2	48.555	27.993	0	1	96.501

Abbreviations used: HOA, human oral absorption; PSA, polar surface area; ROF, rule of five; ROT, rule of three, SAfluorine: solvent-accessible surface area of fluorine atoms.

**Table 3 molecules-28-01593-t003:** Comparative pharmacokinetics and toxicological analyses of studied molecules. Pharmacokinetics analyses were performed with SwissADME, and the toxicological analyses were performed with Pred-hERG and AdmetSAR servers.

PubChem	GI Abs	BBB-p	Pgp-S	CYP1A2-I	CYP2C19-I	CYP2C9-I	CYP2D6-I	CYP3A4-I	AMP
529885	High	Yes	No	No	No	No	No	No	-
27447	High	No	Yes	No	No	No	No	No	-
175002	High	Yes	No	No	No	No	No	No	-
149096	High	No	Yes	No	No	No	No	No	-

Note: Pharmacokinetics analyses were performed with SwissADME, whereas the toxicological analyses were performed with the use of the Pred-hERG and AdmetSAR servers. Abbreviations used: AMP: Ames mutagenicity prediction; BBB-p, blood-brain barrier permeant; GI abs, gastrointestinal absorption; I, inhibitor; S, substrate; -, negative.

**Table 4 molecules-28-01593-t004:** Medicinal chemistry and drug likeness output of test compounds.

PubChem	Lipinski # Violations	Ghose # Violations	Veber # Violations	Egan # Violations	Muegge # Violations	Bioavailability Score	PAINS # Alerts
529885	0	0	0	0	1	0.55	0
27447	0	0	0	1	0	0.55	0
175002	0	0	0	0	0	0.55	0
149096	0	0	0	0	0	0.55	0

**Table 5 molecules-28-01593-t005:** Docking properties of the complexes in this study.

	Compd	Dock Score	MMGBSA	# H-Bond	Pi-Cat	Salt Bridges
**3ILW**	**Co-Ligand**	−5.666	−41.84	GLN277,TRP103,ARG98,PRO124	TRP103	0
	**27447**	−6.13	−34.85	ARG98,TRP108,SER118,GLY 2(120),ASP120	0	ARG98
	**149096**	−4.658	−50.28	THR230,LEU274,HIE 2(280)	0	0
	529885	−4.115	−20.84	ARG98,SER104	0	0
	175002	−2.586	−21.74	TRP103,ARG98	0	0
**6RKU**	**Co-Ligand**	−3.762	−44.57	ALA427	TYR478	0
	**27447**	−5.141	−39.69	ALA421, VAL420	0	0
	**149096**	−3.674	−34.92	GLU381	0	GLU381
	529885	−4.777	−32.94	VAL420, ALA421,ALA427	0	0
	175002	−3.655	−28.4	0	0	0
**1KIJ**	**Co-Ligand**	−7	−58.4	ASP80, ARG135		ARG135
	**27447**	−5.52	−60.81	LYS102, LYS109, GLU49, ASN45	0	ASP48 ARG75
	**149096**	−4.49	−51.88	ASP 72, GLY 76	LYS 109 ARG 75	0
	529885	−5.97	−61.1	ASP48, LYS109	0	0
	175002	−5.86	−15.63	LYS109	0	0

Abbreviations used: 1KIJ, crystal structure of the 43K ATPase domain of *Thermus thermophilus* gyrase B in complex with novobiocin; 3ILW: structure of DNA gyrase subunit A N-terminal domain; 6RKU: *E. coli* DNA gyrase.

**Table 6 molecules-28-01593-t006:** MD simulation properties of the native protein and of the protein–ligand interactions.

Compound	RMSD	RMSF	rGyr	SASA
27447	3.89 ± 0.59	1.33 ± 0.85	3.85 ± 0.12	127.30 ± 26.50
529885	2.87 ± 0.42	1.35 ± 0.85	2.13 ± 0.02	150.99 ± 24.92
Levofloxacin	3.38 ± 0.44	1.57 ± 0.98	3.97 ± 0.05	72.78 ± 22.01
Apo-1KIJ	3.16 ± 0.41	1.29 ± 0.99	-	-

Note: Values are represented as mean ± SEM measured in Å.

## Data Availability

Not applicable.

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
