# Peer review of "Computational and Preclinical Prediction of the Antimicrobial Properties of an Agent Isolated from Monodora myristica: A Novel DNA Gyrase Inhibitor"

_molecules, 2023, doi:10.3390/molecules28041593_

Round 1

Reviewer 1 Report

The present work entitled "Preclinical Prediction of Antimicrobial agent from Monodora myristica extracted oil: A Novel Inhibitor of DNA gyrase in Treatment of Bacterial Cell Proliferation" by Onikanni, Sibuyi et al. computationally explores the properties of the oil extracted from Monodora myristica, showing antimicrobial activity in order to suppress bacteria proliferation, with DNA gyrase as the primary target.

The computational approach employed here involves the Schoeringer package usage to identify a lead compound, considering the binding affinity to compare the efficacy with that of levofloxacin.

The name of the manuscript should include that we are talking about a computational approach, and not an experimental one, as the "new inhibitor for DNA gyrase" was only predicted, and not synthesized, characterized, or biologically tested. This also all over the discussion.

Please, report the abbreviation of MD simulations as MD, not MDS or MDs.

As for the Introduction, focusing on the DNA gyrase inhibition could be nice to explain better the activity of this topoisomerase, maybe with an explication image, as it is the main focus of this work. Same for the inhibitors of DNA gyrase, more inhibitors should be cited, not only levofloxacin (the main standard of this work), reporting their chemical structure in a Figure. The compounds cited at the beginning of page 3 (citations 19,20), could be represented in their chemical structures. for better understanding. The discussion of fatty acids is weak it can be expanded (if crucial for the understanding of the present work), with also more examples.

The writing of the Materials and Methods paragraphs 2.1, 2.2, 2.3 should be written in a more precise way. Please report E. coli in italics.

Compounds in Figure 1 shloud be redrawn in ChemDraw, with a common size and improved quality, showing better "R" and "S".

The parameters in Table 1 should be explained better, with a proper legend under the table.

In page 5, the data do not reveal "no inhibition in Pgp substrate", but show that the compound is not a substrate of Pgp.

Table 5 is a bit confusing, it is not possible to understand what the protein labels refer to (es. 3ILW).

The way of writing the computational results has to be improved with more accuracy and details, correcting the typos and rephrasing.

As for Figures 2-3-4-5-6-7, the 2D representations have to be improved in quality, and the 3D ones is better to have a white background, bold amino acids labels (not readable now), showing better the primary and secondary structure of the protein.

Check better the 2D representation in Figures 8 and 9.

The Discussion includes all the results, and it is well-organized.

The Conclusion has to include which is the main novelty of the present work, and state that it is a computational work and an experimental  effort is needed to confirm the present results.

Author Response

Dear Reviewer, Thanks for your time and contributions. All the points raised has been addressed.

Reviewer 2 Report

The article is very interesting and tackles challenging problems of in silico drug design. However, there are several issues that need to be addressed:
1. In the method section plenty of computational details are missing:
a)How MD was performed? What force field was used (OPLS?)?
b)What software was used to perform the post-MD analysis? It cannot direct to previously used paper and should be stated in the article explicitly.
c)How the system was equilibrated?
d) What was the solvent model?
2. In the article only one trajectory is presented. Did the Authors perform only one trajectory? If yes at least 3 separate trajectories must be performed.
3. From MD simulation MM-GBSA calculation can be performed, how they correlate with the one computed from Prime (from minimization).
Minor comments:
1. The Structures in Figure 1 are rotated, this is incorrect. For example, in Panel D the positive charge which is near nitrogen is located near the lower index of the hydrogen atom confusing with the H2N group. Please provide the correct orientation of the molecules or rotate and generate the indexes properly.
2. Figures 2-7 and 8B are poorly presented. The resolution of those figures is very low, and many labels overlay making them illegible.
3. Conclusion section seems to be unfinished.
4. Many units throughout the manuscript are missing.

Author Response

Dear Reviewer, thank you for your valuable contribution. we have addressed the points raised from the manuscript

Round 2

Reviewer 1 Report

The manuscript can be accepted in the present form, as the Authors addressed all the issues pointed out.

Author Response

Dear Reviewer, see attachment.

Reviewer 2 Report

The authors significantly improved the paper. There is only one major concern that remains valid. The authors performed only one trajectory which is biased by the random number generation of the velocity. The time evolution of that system cannot be treated as separate runs therefore SEM is not bias free estimator. Authors must perform at least three separate runs (as only then can ensure the independent unbiased velocity generation).

Once the Authors perform 3 independent trajectories the article can be accepted.

Author Response

Dear Reviewer, See attachment.
